# The Role of Myeloid-Derived Suppressor Cells (MDSCs) in Graft-versus-Host Disease (GVHD)

**DOI:** 10.3390/jcm10102050

**Published:** 2021-05-11

**Authors:** Christos Demosthenous, Ioanna Sakellari, Vassiliki Douka, Penelope Georgia Papayanni, Achilles Anagnostopoulos, Eleni Gavriilaki

**Affiliations:** Hematology Department—BMT Unit, G Papanicolaou Hospital, 57010 Thessaloniki, Greece; christosde@msn.com (C.D.); ioannamarilena@gmail.com (I.S.); vassiliki.douka@gmail.com (V.D.); papagianni.pinelopi@gmail.com (P.G.P.); achanagh@gmail.com (A.A.)

**Keywords:** NS cells, MDSCs, allo-HCT, GVHD, GVL

## Abstract

Background: Myeloid-derived suppressor cells (MDSCs) are implicated in the complex interplay involving graft-versus-leukemia (GVL) effects and graft-versus-host disease (GVHD) after allogeneic hematopoietic stem cell transplantation (allo-HCT) in hematologic malignancies. Methods: A review of literature through PubMed was undertaken to summarize the published evidence on the pathophysiology and clinical implications of MDSCs in allo-HCT. Literature sources published in English since 1978 were searched, using the terms Natural Suppressor (NS) cells, MDSCs, GVHD, and allo-HCT. Results: In vivo studies demonstrated that MDSCs derived from mobilization protocols could strongly suppress allo-responses mediated by T cells and enhance T-Reg activity, thus inhibiting GVHD toxicity. However, the influence of MDSCs on the GVL effect is not fully defined. Conclusions: The induction or maintenance of MDSC suppressive function would be advantageous in suppressing inflammation associated with GVHD. Pathways involved in MDSC metabolism and the inflammasome signaling are a promising field of study to elucidate the function of MDSCs in the pathogenesis of GVHD and translate these findings to a clinical setting.

## 1. Introduction

Allogeneic hematopoietic cell transplantation (allo-HCT) is currently considered the only potentially curative option for various life-threatening malignant and benign hematologic conditions. However, allo-HCT poses two major limitations to its success; immune reactions of the (a) host versus graft (HVG), which may lead to engraftment failure, and (b) graft versus host (GVH), leading to the homonymous disease (GVHD). As the quest for the expansion of donor pool with alternative transplantations (haploidentical, umbilical cord blood, or mismatched unrelated) continues, acute and/or chronic GVHD (aGVHD and cGVHD) are leading causes of morbidity and mortality [1,2].

The backbone of conventional treatment for aGVHD relies on the use of corticosteroids [3]. Corticosteroids, as the gold standard of initial therapy, result in diverse complete responses (25–69%) [4]. As aGVHD severity increases, so does mortality, which is indicated by mortality rates reaching 95% in grade III-IV aGVHD. Despite the research community’s efforts to establish novel targeted strategies against GVHD, no optimal treatment regimen has been described [5,6,7]. Guidelines from the European Group for Blood and Marrow Transplantation (EBMT) and the European LeukemiaNet (ELN) do not propose a standard second-line treatment for acute or chronic GVHD [8]. Similar are the recommendations from the American Society of Blood and Marrow Transplantation (ASBMT) for aGVHD [9]. Suggested alternative treatment options consist of various immunomodulatory treatments, including extracorporeal photopheresis (ECP), mycophenolate mofetil (MMF), anti-tumor necrosis factor a (anti-TNF-α) antibodies, mammalian/mechanistic target of rapamycin (mTOR) inhibitors, or interleukin-2 receptor (IL2R) antibodies. Novel biologic treatments have also been recently incorporated in the treatment of GVHD, such as ruxolitinib or ibrutinib [10]. Ruxolitinib was recently FDA-approved for the treatment of steroid-refractory aGVHD, based on REACH-1 and REACH-2 trial results that showed overall response rates of 55–62%, which are significantly greater than those observed with former “standard of care” options [5,6]. Additionally, ibrutinib has demonstrated efficacy in a prospective trial of steroid-refractory chronic GVHD but without the involvement of a comparator group [7]. Nevertheless, long-term efficacy and safety of these agents remain to be determined. Overall, all patients undergoing allo-HCT should receive prophylaxis to mitigate or prevent GVHD. Choices of regimens have been proposed for aGVHD prophylaxis, including methotrexate (MTX) and post-transplant cyclophosphamide (PTCy) [10,11].

Myeloid suppressor cells are implicated in the complex interplay involving graft-versus-leukemia (GVL) effects, GVHD, and hematological malignancies. In the microenvironment of hematological malignancies, several studies have highlighted the role of myeloid-derived suppressor cells (MDSCs) [12,13]. The pathogenic role of MDSCs in solid malignancies has been reviewed by others [14,15,16,17]. Recently, the multiple roles of MDSCs have also been reviewed in hematological malignancies [13,18] and diseases in general [13]. As reported, the intensification of anti-tumor responses by inhibiting negative immune regulators is a more common mechanism in chronic myelogenous leukemia, lymphoma, high-risk myelodysplastic syndrome (MDS), and multiple myeloma, but not in leukemia [13,18]. Nowadays, MDSCs are attracting interest in the context of allo-HCT and aGVHD due to their role as immunoregulators and alloreactive T cell suppressors.

In this review, we will try to shine a light on (a) key functions of MDSCs, (b) the role of MDSCs after allo-HCT on GVHD development, and (c) the implications of targeting MDSCs.

## 2. Key Functions of MDSCs

MDSCs are a diverse population of naturally occurring immature myeloid cells characterized by their capacity to suppress responses mediated by both innate and adaptive immunity. MDSCs arise under chronic inflammatory conditions such as malignancy, infection, autoimmune diseases, trauma, GVHD, and so on. Although phenotypically similar to mature myeloid cells, they pose distinct genomic and biochemical profiles and function [19,20]. In the presence of tumor, relatively immature and pathologically activated MDSCs with dominant immunosuppressive activity are known to accumulate in the lymph nodes (LNs), spleen, and liver. MDSCs have the ability to modulate malignancy by augmenting tumor cell survival, angiogenesis, tissue invasion, and hematogenous spread [17,21].

In 1978, Bennett and colleagues were the first to demonstrate the presence of natural suppressor, myeloid cells in bone marrow and spleen of irradiated mice with the ability to suppress T cell activity [22]. The potential role of this cell population expressing a “null” surface phenotype against GVHD, was recognized later, in the mid-1980s, in several other studies involving mice [23,24,25]. Labeled as natural suppressor cells (NS) [23], these cells showed suppressive activity in a non-antigen specific nor major histocompatibility complex (MHC) restricted manner [23,24,25,26]. The field began changing with the identification of evidence of unique regulatory function and surface phenotype of these cells [27]. NS cells were later identified as MDSCs [28]. In murine models, MDSCs historically were classified as cells expressing simultaneously Gr-1 and CD11b antigens. Although originally useful in recognizing MDSCs, the use of this criterion is no longer sufficient, as subpopulations have been recognized: polymorphonuclear or granulocytic MDSCs (PMN/G−MDSCs, CD11b+ Ly6G+ Ly6Clow) and monocytic MDSCs (M-MDSCs, CD11b+ Ly6G− Ly6Chigh) [20]. Three different groups of human MDSCs have been identified based on their cellular markers, as determined by flow cytometry. Polymorphonuclear or granulocytic-MDSCs (PMN/G-MDSCs, CD11b+ CD15+ CD14− or CD11b+ CD14− CD66+ cells), monocytic-MDSCs (M-MDSCs, CD11b+ CD14+ HLA-DRlow/− CD15− cells) and a subset of more immature progenitors called early-stage MDSCs, (eMDSCs, lineage-negative (CD3/14/15/19/56)/HLA-DR−/CD33+) [13,20]. Additional populations have been described by several other groups, such as tumor-induced programmed death-ligand 1 (PD-L1)+ CD33+ CD11b+ HLA-DR− MDSCs, [29] CXCR1+ CD14+ CD15− HLA-DR−/low, [30], and secreted protein acidic and rich in cysteine (SPARC)-positive MDSCs [31].

As mentioned, MDSCs arise under chronic inflammatory stimuli. A two-signal model was described to explain the subsequent differentiation of myeloid suppressor cells [13,32]. A first group of signals involving macrophage colony-stimulating factor (M-CSF), granulocyte colony-stimulating factor (G-CSF), and granulocyte–macrophage colony-stimulating factor (GM-CSF) results in initial generation of MDSCs in the bone marrow, in a similar manner to mature myeloid cells. The second phase is driven by proinflammatory cytokines produced by inflammatory and tumor microenvironment. Interferon (IFN)-γ, interleukin (IL)-4, IL-6, and IL-13 finally distinguish MDSCs, according to special gene expression profiles from mature neutrophils and monocytes in healthy donors [16,32].

Available data nowadays support the notion that these gene and protein expression profiles allow MDSCs to utilize a number of mechanisms to suppress T cell responses, as demonstrated both in vitro and in vivo. The hyperproduction of inducible nitric oxidase (iNOS), arginase 1 (ARG1), [33] reactive oxygen species (ROS) [34], secretion of TGF-β [35], upregulation of cyclo-oxygenase 2 (COX2) and prostaglandin E2 (PGE2) [36], as well as enhanced activity of indoleamine 2,3-dioxygenase (2,3-IDO) directly inhibit T cell, B cell, and NK cell activation and expansion [37,38,39]. Furthermore, MDSCs drive immune suppression through the activation of regulatory T cells (T Regs) and upregulation of immunosuppressive cytokines such as TGFβ and IL-10 [40,41,42,43]. It is now evident that G-MDSCs and M-MDSCs utilize discrete mechanisms for immunoregulation [44]. Specifically, G-MDSCs possess the ability to suppress T cells in an antigen-specific manner preferentially using ROS, ARG1, and PGE2 [19]. M-MDSCs, on the other hand, inhibit T cells both in an antigen-specific and nonspecific mechanisms utilizing NO and immunosuppressive cytokines [19].

## 3. GVHD Pathophysiology and Implications for MDSCs

The pathophysiology of aGVHD has been connected to a 3-phase process: (1) *initial tissue damage* from the conditioning regimen, which activates host antigen-presenting cells (APC) by danger-associated molecular patterns (DAMPs) and pathogen-associated molecular patterns (PAMPs), (2) *afferent phase* characterized by the stimulation and proliferation of donor T cells in response to alloantigen expressed either on host APCs, labeled as direct antigen presentation, or on donor APCs, labeled as indirect presentation, and the (3) *effector phase* represented by generated donor T cell-mediated cytotoxic damage against host cells through Fas–Fas ligand interaction, perforin–granzyme, and TNF-α [45].

Through their capacity to attack the recipient’s tissues, donor alloreactive T cells are considered a potential target to suppress aGVHD reactions and reduce organ injury. Recently, there has been increasing interest in the contribution of donor MDSCs on GVHD management due to their immunosuppressive effects on alloreactive T cell priming and expansion and induction of T Regs (Table 1). As it has been demonstrated, MDSCs can be mobilized from normal BM in a relatively short amount of time and have the ability to inhibit GVHD as well as allograft rejection efficiently [46].

More than two decades ago, Mielcarek et al. showed that mononuclear cells from mobilized blood obtained from healthy donors after G-CSF initiation were less responsive (31.5% ± 9.2% response, *p* = 0.003) compared to mononuclear cells collected from the peripheral blood before the administration of G-CSF. This study also demonstrated that G-CSF-mobilized peripheral blood CD14+ mononuclear cells had the ability to suppress alloantigen-induced proliferation of CD4+ T cells by more than 50% [61]. An inverse relationship between suppressor cell activity and CD4 T cell apoptosis was further documented following autologous stem cell transplantation [62]. Subsequent studies by Luyckx et al. aimed to characterize these cells. Flow cytometry on individual peripheral blood samples from six different G-CSF-treated PBSC-donors identified a MDSC population (Lin−, HLA-DR−, CD11b+), reaching a median value of 90% (range 83.5–94.5%) among a CD45+ cell population, whereas in the control individuals, this was only 35% (range 21.7–58.0%). The expanded MDSC population comprised monocytic (CD33high, CD14high, SSClow, CD15−) and granulocytic (CD33int, CD14low, SSChigh, CD15high) subpopulations with the ability to suppress T cell alloreactivity [12].

### 3.1. Early Studies in Murine Models

Early murine studies identified enhanced activity of lymphocyte suppressor myeloid cells and concomitant appearance of these cells in the spleen of thymectomized, irradiated, bone marrow-reconstituted mice after systemic bacillus Calmette–Guerin (BCG) administration [22]. Subsequent research confirmed transient accumulation of suppressor cells, deficient of surface markers for macrophages, NK cells, B cells, or T cells, in the lymphoid tissues of neonatal or irradiated mice [23,24]. To further elucidate the function of these “null” cells or natural suppressor cells (NS), more studies were performed in murine GVHD models. These cases involved rapidly proliferating stem cell populations, including NS cells capable of suppressing alloreactive T cells in an antigen-independent manner [25], raising questions about whether an implication in GVHD could exist [25,63]. NS cells were demonstrated to arise during the first weeks after bone marrow transplantation in lethally irradiated mice [64]. NS cells were also identified in normal bone marrow with comparable characteristics to that in spleens of early bone marrow transplant recipients [65], providing at that time a possible source for obtaining NS-enriched populations for adoptive transfer studies after in vitro expansion [65]. The myeloid origin of NS cells was confirmed later, as they were quantified as progenitors of monocyte lineage by colony-forming assays in soft agar [66] before the term “MDSC” was introduced [28]. More recent murine studies showed that MDSCs (Gr-1+ and CD11b+ population) seem to accumulate during the first 2 weeks after allo-HCT. This phenomenon is probably driven by the proinflammatory cytokine storm mediated by the preparative regimen. Involved cytokines include IFN-γ, G-CSF, IL-1β, and IL-6, creating an ideal stroma for accumulation and activation [67,68]. In the absence of aGVHD, MDSC percentage returns to basal level by week 6, parallel to the decline of cytokine levels. On the other hand, the development of GVHD is related to a constant enhancement of MDSC in a severity-dependent manner (*p* < 0.05, Student’s *t* test) either due to stimulation by cytokines or due to interaction with the bone marrow and spleen microenvironments [49].

Interestingly, the treatment of donor mice with CpG and incomplete Freund’s adjuvant (IFA) increased MDSCs (CD11b+ Gr-1+) that abrogated T cell alloreactivity in vitro and GVHD in vivo. This model also demonstrated increased levels of IL-6 and IL-10 after treatment with CpG; however, a direct association of increased cytokine levels with T cell anergy has not been demonstrated [52]. On the other hand, interleukin-13 (IL-13) production enabled a Ly6C+, CD11b+, Arg1+ (MDSC-IL13) population, promoting metabolic stress and T cell dysfunction [54]. Messmann and colleagues reported that MDSCs generated in vitro after culture of bone marrow cells with G-CSF and GM-CSF inhibited GVHD-induced death by 80%, preferentially by inducible nitric oxide synthase (iNOS) and attenuated histologic GVHD through Th2 induction. In contrast, the anti-tumor cytotoxicity of alloantigen-specific T cells was maintained (*p* ≤ 0.05, Student’s *t*-test) [48]. The immunosuppressive function of MDSCs, obtained after administration of G-CSF on GVHD models, has been elucidated further by other in vivo studies [51]. Wang and colleagues acquainted the ability of G-CSF to proliferate MDSCs and equip HLA-DR−/low, CD33+, CD16− eMDSCs with regulatory properties against the expansion of autologous CD3+ T cells in a TGFβ-dependent manner. Meanwhile, eMDSCs managed to promote T Reg upregulation and polarization of Th1/Th17 cells to Th2 cells [51]. These findings were translated into an attenuation of aGVHD and increased survival [51]. In studies performed by Messmann et al., skewing of T cells toward Th2 cells and attenuation of intestinal and cutaneous GVHD manifestations were independently of MHC class I expression and antigen presentation [48]. Interestingly, allogeneic T cell proliferation and homing were not influenced under these circumstances, thus maintaining anti-tumor activity [48]. The previous year, a novel approach utilizing systemic infusion of combined donor-derived-MDSCs and T Regs obtained after G-CSF has successfully demonstrated ameliorated inflammation in an aGVHD mouse model [60]. According to histopathologic analysis, the extent of tissue inflammation and lymphocyte infiltration were significantly reduced in all primary target organs of aGVHD, such as the skin, intestine, and liver [60].

To pinpoint the mechanism by which MDSCs protect from GVHD, Joo and colleagues evaluated whether indoleamine 2,3-dioxygenase (IDO) is implicated in T cell suppression, thus mimicking immunomodulation observed between maternal T cells and fetal tissue during pregnancy. Apart from confirming the regulatory effects of G-CSF in allo-HCT, they showed that this impact was mediated by IFN-γ-induced 2,3-IDO functional activity [53]. Moreover, the suppressive functions of MDSCs are assumed to be reactive oxygen species (ROS) dependent [49].

Further studies validated the immunosuppressive role of ARG1 after HCT and inhibition of alloreactive T cell responses [49,54]. L-arginine (l-Arg) is a nonessential amino acid with a central role in T cell proliferation and function. l-Arg is catabolized by ARG1, an enzyme produced by MDCSs, resulting in reversed T cell function by suppressing T cell expression of the CD3ζ chain and the cell-cycle regulators, cyclin D3, and cyclin-dependent kinase 4 [69].

As previously mentioned, iNOS is involved in MDSC-related immunoregulation. NOS is a known antiproliferative for T lymphocytes, B lymphocytes, and NK cells [39,70]. The leading MDSC group capable of producing NO is believed to be the monocyte-MDSC subset (CD11b+ Ly6Chigh Ly6Glow) and, more specifically, a highly immunosuppressive subpopulation expressing CD34+. The local production of NO, in response to IFN-γ produced by activated T cells, was able to reduce T cells activation and proliferation and protect from GVHD in vivo [55].

Although the potency of MDSCs generated ex vivo in aGVHD was well documented, in vivo activity was limited as MDSCs lost their suppressive ability after setting into a highly inflammatory environment within the HCT recipient. Previous studies in aGVHD emphasized the critical role of NLRP3 inflammasome pathway in MDSC alloimmune stimulation [71]. This limitation was related to cell-intrinsic inflammasome upregulation, production of inflammatory mediators, and myeloid differentiation [46]. The inflammasome is an intracellular multiprotein complex that controls the induction of inflammatory caspases such as caspase-1 and -11 [71]. To elucidate the environmental and intrinsic mechanisms of MDSC activation, Koehn et al. evaluated the implication of the NLRP3 inflammasome pathway in an aGVHD major histocompatibility mismatch murine model [47]. Factors related to inflammasome activation are produced during allo-HCT (e.g., preparative regimen, aGVHD), including intestinal release of bacterial products and danger-associated molecules from dying cells that translocate into the internal milieu [32]. In their studies, Koehn and colleagues showed that preparative regimen-induced adenosine triphosphate (ATP) relocation is a main driver of MDSC dysfunction mediated by ATP receptor (P2x7R) engagement and NLR pyrin family domain 3 (NLRP3) inflammasome stimulation. Taking it one step further, they also revealed that P2x7 knockout or inhibition of ATP association with P2x7R receptor inhibited inflammasome activation. The latter was exhibited with extracellular ATP exhaustion via apyrase and pharmacologically via treatment with A-438079, which is a highly selective P2x7R inhibitor [47].

The Toll-Like Receptor 4 (TLR4)–Myeloid Differentiation primary response gene 88 (MyD88) pathway is also contributing to insufficient expansion of donor MDSCs and initiation of aGVHD [56,72,73], whereas pharmacological induction of TLR4 aggravates GVHD lethality [74]. In the context of allo-HCT, the activation of TLR4/MyD88 results from intestinal microbiota products, such as lipopolysaccharides (LPS), which are released after intestinal injury from the conditioning. In T cell-depleted BM murine transplant, the expression of MyD88, but not lack of it, was essential for protection against fatal intestinal aGVHD as observed after repeated LPS injections. MyD88 favored the presence of CD11b+ Gr-1+ cells in target organs, whereas at the same time, the degree of intestinal T cell infiltration was inferior [56].

mTOR pathway inhibitors, such as sirolimus and rapamycin, are being increasingly used in the prophylaxis and management of aGVHD. Their multiple activities include immunosuppressive actions through T cell suppression while promoting T Regs, inhibition of antigen presentation and dendritic cell maturation, antifibrotic properties, antineoplastic, and antiviral activities [75]. In vivo studies showed that rapamycin was able to induce G-MDSCs accumulation with an enhanced immunosuppressive role in the presence of aGVHD, via upregulation of ARG1 and iNOS and induction of regulatory T cells. Graft-versus-tumor effect was maintained [57,58].

Another mechanism through which MDSCs are involved in aGVHD is the upregulation of T Regs. The cytokine-driven polarization of T cells from Th1/Th17 to Th2 secondary to increased IL-4/INF-γ and IL-4/IL-17 ratio has been proposed [51]. Il-10, produced by granulocyte-MDSCs, might also have a central role in this. As demonstrated, the treatment of donor mice with ProGP-1 and G-CSF expanded a CD11bhighGr-1low population that induced IL-10-producing regulatory T cells, likely via indirect presentation of host antigens within the context of donor MHC class II, and prevented GVHD, while the GVL effect was preserved (*p* < 0.0001, Kaplan–Meier and compared by log-rank analysis) [43]. Moreover, MDSCs upregulate T Regs through the enhanced expression of ligands for T cell co-stimulatory molecule NKG2D, such as RAE-1 and MULT-1 [50].

While many studies delineated the role of MDSCs on aGVHD, the knowledge concerning that association with cGVHD is limited. Recently, Lim et al. showed in a preclinical model that ex vivo-generated human cord blood MDSCs (CD14+, HLA-DRlow, CD11b+, CD33+) limited clinical and pathologic cGVHD severity by alleviating thymic damage and attenuating Th 17 and Th 2 differentiation, proposing a possible therapeutic strategy for the clinical application of MDSC infusion [76].

### 3.2. Studies in Humans

Before the introduction of the term “MDSCs”, studies documented an increase of NS cells in the apheresis products following mobilization compared to bone marrow and cord blood products [77,78,79]. In particular, Mills and colleagues demonstrated an increased frequency of natural suppressor cells in GM-CSF-mobilized products and bone marrow cells of patients with NHL, compared to cord blood products and unmobilized apheresed mononuclear cells from healthy volunteers [77]. Moreover, Talmadge et al. were the first to report high levels of suppressor cell activity in the peripheral blood of the patients after transplantation [78]. These studies involved patients with solid or hematological malignancies, and all documented an inverse relationship between NS cells and T cell number and function [77,78]. Concerning products derived from healthy donors and besides the anticipated recruitment of hemopoietic progenitors, rhG-CSF also managed to produce an unexpected modification of lymphocyte subsets [79]. In accordance, more recent data provided evidence that cell populations representing M-MDSC and G-MDSCs increase in healthy donors’ peripheral blood during G-CSF mobilization [12,80], irrespective of age and sex [77]. Research confirmed the inverse correlation between the incidence of aGVHD and the presence of M-MDSCs [55,80], and eMDSCs in the graft [51]. A lower incidence of aGVHD was observed both in the haploidentical and HLA-matched allo-HCT setting [81]. Furthermore, Lv and colleagues correlated negatively the presence of MDSC with cGVHD, documenting no significant effect on relapse and survival [81]. Individuals receiving higher numbers of G-MDSCs and M-MDSCs in grafts displayed a lower incidence of grade II-IV aGVHD and severe cGVHD after haploidentical HCT [81]. The accumulation of M-MDSCs in patients after allo-HCT, especially during higher grade aGVHD, was associated with the suppression of CD3ζ-chain expression on T cells via 2,3-IDO [38]. Although the proportion of M-MDSCs cells correlated significantly with G-CSF administration in donors [12,80], G-CSF administration in patients at any time-point after allo-HCT did not influence CD14+HLA-DRlow frequency [38]. However, retrospective data showed improved anti-leukemic effect after administering G-CSF–donor lymphocyte infusions (G-CSF-DLI) compared to regular–DLIs in relapsing patients post allo-HCT. G-CSF-DLI were enriched with G-MDSCs and M-MDSCs and, remarkably, they did not increase the cumulative incidence of GVHD [82].

The role of ECP as an alternative treatment approach in GVHD has been well established [83,84,85]. The infusion of ECP-treated leukocytes in steroid-refractory GVHD has been demonstrated to mobilize P-MDCS [86] or CD33+CD11b+ MDSC subsets [87]. MDSC levels were particularly enhanced in aGVHD patients compared to patients with cGVHD, and they seem to possess a vital role in the immunomodulatory modality of ECP [87].

Primary target organs of aGVHD include the skin, the liver, and the intestine [3,88]. Unfortunately, limited information could be recovered during the review of the literature regarding the association between GVHD manifestation and MDSCs. Of note, a prospective clinical study by Vendramin et al. demonstrated that G-CSF induced the expansion of M-MDSCs capable of abrogating aGVHD by reducing tissue damage and inflammation in all target organs [80]. More data were obtained from studies involving the administration of G-CSF mobilized MDSCs in xenogeneic models of GVHD. A study from D’Aveni et al. showed lower GVHD histopathological scores, mainly in the colon, in humanized mouse models after the infusion of G-CSF-mobilized CD34+ monocytes [55]. A similar effect was demonstrated by Wang and colleagues, who showed that G-CSF mobilized HLA-DR−/lowCD33+CD16− cells resulted in improved histopathological score in hepatic and intestinal tissues [51].

M-MDSCs and invariant natural killer (iNKT) cells were shown to expand shortly after allo-HCT [89]. The delayed recovery of both M-MDSCs and iNKT cells following transplantation was associated with an increased occurrence of grade III-IV aGVHD. Nevertheless, the combination of lower M-MDSCs and higher iNKT cells correlated with enhancement of GVL effect and decreased the rate of leukemia relapse [89]. A major obstacle facing the field of GVHD management through suppressor cells is deregulation of the immune system, delayed reconstitution, and increased susceptibility to infections. The conditioning regimen in conjunction with stem cell infusion and subsequent tissue damage creates an inflammatory milieu that might induce an exaggerated expansion of M-MDSCs [90]. In such circumstances, increased M-MDSC levels predict for higher non-relapse mortality [90].

The experimental information from animal models and observations from patients undergoing allo-HCT indicates that MDSCs illustrate a promising tool for preventing and managing GVHD in the clinic. On the other hand, limited data were derived from human studies. Generating MDSCs from human is difficult due to low concentration. More than one month of cultivation is required to produce sufficient MDSCs which might be too long for their application as a treatment for severe acute GVHD, whereas the ideal method for their processing has not yet been identified [91].

## 4. MDSCs as Diagnostic or Therapeutic Targets in GVHD

In the clinical setting, the prediction of the likelihood of GVHD is critical from a therapeutic perspective. As it was shown before, the calculated T cell/MDSC (HLA-DR-CD11b+ CD33+) ratios in peripheral blood mononuclear cells among 55 patients who experienced allo-HCT with a myeloablative conditioning regimen at the time of engraftment were significantly higher in patients that further developed intestinal GVHD ≥ grade 2 (*p* = 0.03) [56]. Broad-spectrum antibiotic use (e.g., carbapenem) resulting in loss of microbiota diversity, reduced recovery of M- and eMDSCs, and an increased occurrence of intestinal GVHD [92]. This link between MDSC and intestinal GVHD might be explained by the higher incidence of GVHD and mortality in allo-HCT recipients with loss of microbiota diversity [93]. Moreover, M-MDSCs and iNKT cells in patient’s peripheral blood early after allo-HCT can be attractive biomarkers to predict transplantation outcomes including leukemia relapse and aGVHD [89]. In detail, the combination of higher frequency of M-MDSCs and lower frequency of iNKT cell revealed increased relapse probability compared to those with lower frequency of M-MDSCs and higher frequency of iNKT [20.9% (95% CI, 12.4–52.9) vs. 2.9% (95% CI, 0.2–12.6), *p* = 0.011] [89]. On the other hand, combination of higher frequencies of M-MDSCs and iNKT cells was associated with decreased incidence of grade III~IV aGVHD compared to those of lower frequencies of M-MDSCs and iNKT cells (2.8% (95% CI, 0.2–12.6) vs. 31.6% (95% CI, 12.4–52.9), *p* = 0.002) [89].

Considering their significant implications in GVHD, there is growing interest of developing agents targeting pathways where MDSCs are involved. The immunoregulatory properties of MDSCs have been demonstrated by several studies performed at different time points of a transplantation procedure: in the graft and in the donor along the post-transplantation period (Figure 1). Donor T cells that acquire the ability to attack the recipient’s tissue have a crucial role in the development of aGVHD; thus, the most efficacious approaches to prevent lethality are those that hamper T cell responses early after transplantation when inflammation is escalating driven by alloreactive T cells [94].

However, the hyper-responsiveness of MDSCs makes them difficult to track over time along the course of a transplanted patient. When exposed to inflammatory conditions, and in contrast to suppressed lymphocytic lineage, immature myeloid cells undergo emergency granulopoiesis [95] driven by cell-intrinsic inflammasome activation and the release of inflammatory mediators [46]. An initiating event in downstream NLPR3 inflammasome activation is the release of ATP and other DAMP molecules after the damage of tissues induced by the conditioning regimen. The binding of ATP to the P2X2 receptor and subsequent canonical inflammasome activation launches a cascade of proinflammatory effects that nourish aGVHD [96,97]. In this concept, the use of small-molecule inhibitors of inflammasome activation being developed for other inflammatory conditions may be suitable for MDSC after allo-HCT [98,99]. Notably, Koehn and colleagues demonstrated in several studies that the pharmacologic inhibition of P2X2 receptor and NLPR3 inhibited aGVHD and increased overall survival [46,47]. The production of mitochondrial ROS during the preparative regimen represents another stimulus for NLPR3 inflammasome activation [100]; however, a direct association with redox homeostasis in MDSC is not yet clear.

The synergy between T Regs and MDSCs is another promising direction that warrants further investigation to diminish GVHD without a negative effect on GVL. In the presence of IFN-γ, MDSCs secrete IL-10 and TGFβ, activating thus T Regs (*p* < 0.01, ANOVA) [41]. In addition, under inflammatory conditions, T Regs may promote immunosuppression by the differentiation of MDSC toward M-MDSCs (*p* < 0.01, ANOVA and Tukey post-test) [101]. The use of rapamycin is shown to be feasible in aGVHD management when the immunosuppressive capacity of MDSCs needs to be strengthened without impairing T cell-mediated fitness [57,58]. Supported by these data, it is logical for someone to consider that the adoptive transfer of a product containing both mobilized MDSCs and T Regs may provide greater benefit for achieving immune tolerance than either alone. Indeed, Park et al. showed that combined therapy resulted in suppressive activity in vivo and effectively ameliorated GVHD evolution [60]. However, higher percentages of T Regs in patients were also associated with the amelioration of GVL effect and inadequate response to donor lymphocytes [102,103]. Moreover, increased T Regs were also encountered more frequently in patients with fatal infections [103], displaying the multifactorial role of T Regs on overall survival (OS) after allo-HCT. In the latter case, the presence of MDSCs did not seem to influence GVL outcome [103]. The clinical success to date is a complicated issue, and previous G-CSF, origin, phenotype, dose, and timing of delivery of MDSCs and T Regs might influence therapeutic efficacy.

The treatment of allo-HCT murine models with galectin-9 (Gal-9), a β-galactoside-binding soluble lectin family member, increased G-MDSC frequencies that showed enhanced expression of Arg-1 and iNOS and inhibited T cell proliferation, ameliorating thus the severity of aGVHD [59]. Supported by these data and due to their concentration-dependent nature, Gal-9 represents a potential prognostic biomarker of aGVHD, reflecting the presence and immunosuppressive activity of G-MDSCs.

Finally, a better understanding of the role of microRNAs (miRs) in MDSC and subsequent T cell alloreactivity can be explored in terms of enhanced MDSC suppression and GVHD inhibition. miRs represent a family of small, non-coding RNAs that function as the significant endogenous triggers for RNA interference [104], whereas miR-155 has been implicated in NLRP3 inflammasome activation and CD8+ T cell immunity [105]. In support of this notion, Chen et al. demonstrated recently with studies in murine models that knocking out miR-155 from dendritic cells was associated with less severe GVHD through reduced migration and defective inflammasome activation [106].

## 5. Conclusions

In conclusion, MDSCs derived from mobilization protocols can strongly suppress allo-responses mediated primarily by T cells and to a lesser degree by B cells and NK cells. At the same time, MDSCs promote T Reg expansion. Transplants using mobilized peripheral blood stem cell products infuse approximately 10-fold more T cells than that found in bone marrow products. Nevertheless, the absence of GVHD in these patients has been associated with the infusion of MDSCs in these products [107]. The induction or maintenance of MDSC suppressive function would be beneficial in inhibiting inflammation associated with GVHD. Data from several studies propose the administration of ex vivo constructed MDSCs or in vivo interventions to achieve a desired suppressive state in vivo.

However, pathways involved in MDSC metabolism and the inflammasome signaling should be studied thoroughly in order to gain more knowledge regarding the applicability of these practices in the clinical setting. Furthermore, translational studies are highly needed in this field to confirm the role of MDSC in the pathogenesis of GVHD. While entering the era of CAR-T cell and other upcoming cellular therapies [108], appreciating the interactions between effector immune cells and MDSCs in order to be able to define the ideal conditions for harvesting is of paramount importance.

## Figures and Tables

**Figure 1 jcm-10-02050-f001:**
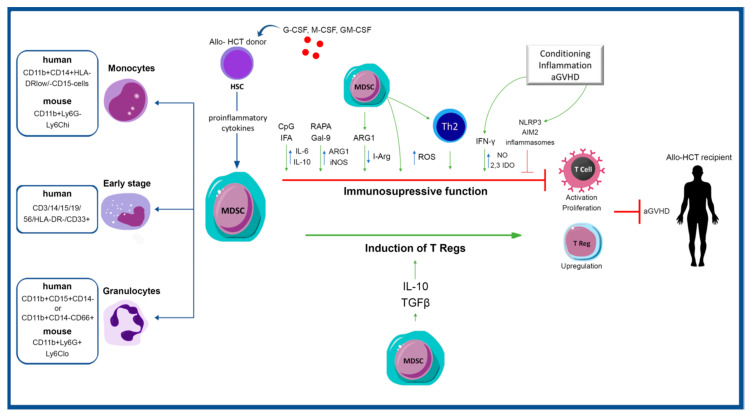
Implications of MDSCs in GVHD. Myeloid-derived suppressor cells (MDSCs) are generated under inflammatory conditions. Inflammatory factors that induce MDSC recruitment and expansion include GM-CSF, G-CSF, M-CSF, and proinflammatory cytokines. Three different subsets of MDSCs have been identified according to their cellular markers; granulocytes, monocytes, and early stage MDSCs. The role of MDSC in the pathogenesis of GVHD would be beneficial through two mechanisms: (1) Immunosuppressive activity of MDSCs by regulating the secretion of various factors, among which ARG1, iNOS, and IDO; (2) Induction and expansion of T Regs. GM-CSF, granulocyte–macrophage colony-stimulating factor; G-CSF, granulocyte colony-stimulating factor; M-CSF, macrophage colony-stimulating factor; HSC, hematopoietic stem cells; HCT, hematopoietic cell transplantation; IFA, incomplete Freund’s adjuvant; RAPA, rapamycin; Gal-9, galectin-9; I-Arg, L-arginine; IFN-γ, interferon gamma; NO, nitric oxide; NLRP3, NLR pyrin family domain 3; AIM2, absent in melanoma 2; aGVHD, acute Graft-versus-Host Disease; ARG1, arginase 1; iNOS, inducible nitric oxide synthase; IDO, indoleamine 2,3-dioxygenase; ROS, reactive oxygen species; TGFβ, transforming growth factor beta; IL-6, IL-10, interleukin-6, interleukin-10.

**Table 1 jcm-10-02050-t001:** Summary of clinical findings and mechanisms for each MDSC immunophenotype.

MDSC Immunophenotype	Clinical Finding	Mechanism	Reference
CD11clow, MHCIIlow, F4/80int	Exposure to inflammasome-stimulating mediators negates the suppressive function of cultured murine and human-derived MDSCs	MDSC-IL13 were activated for NLRP3 or AIM2 inflammasomes using either LPS plus ATP or LPS plus poly(dT) transfection	Koehn et al. [46]
Murine MDSC-IL13s	Inhibition of the inflammasome pathway resulted in maintained MDSC function and improved survival after HSCT in the aGVHD model	Inflammasome activation was reduced via P2x7 knockout (KO) or suppression of ATP binding to the receptor (exhibited with extracellular ATP depletion via apyrase and pharmacologically via administration of A-438079, a highly specific P2x7R inhibitor)	Koehn et al. [47]
G-MDSCs expressing CD11b+Gr-1+, Ly-6ClowLy-6G+ and M-MDSCs expressing CD11b+Gr-1+, Ly-6ChighLy-6G−	MDSCs prevented GVHD-induced death and diminished histologic GVHD	MDSCs induce Th2, while anti-tumor cytotoxicity of alloantigen-specific T cells was preserved	Messmann et al. [48]
H-2Kb+CD11b+Gr-1+	Addition of functional MDSCs in donor graft-attenuated GVHD, while the removal of MDSCs in vivo exacerbated GVHD. MDSCs derived from recipients with GVHD demonstrated induced suppressive potency compared with those from recipients without GVHD. Tumor relapse allowed progressive accumulation of MDSCs in the peripheral blood and spleens of recipients after allo-HCT. Thus, monitoring blood MDSCs may predict relapse	MDSCs suppress alloreactive T cell responses	Wang et al. [49]
CD115+Gr-1+F4/80+	MDSCs effectively attenuated GVHD but did not significantly compromise GVL effects	MDSC demonstrated cytolytic activities against allogeneic leukemia cells via induction of NKG2D+ CD8 T cells, whereas suppressed GVHD through upregulation of T Regs	Zhang et al. [50]
HLA-DR−/lowCD33+CD16- cells	eMDSCs prevented GVHD in humanized mouse model and suppressed the occurrence of grade II-IV aGVHD in allo-HCT patients	eMDSCs are implicated in T Reg upregulation and polarization of T cells from Th1/Th17 to Th2	Wang et al. [51]
CD11bhighGr-1low	MDSCs induce IL-10-producing T Reg that inhibit GVHD through MHC class II restriction	Indirect presentation of host (H-2d) peptides throughMHC class II donor molecules	McDonald et al. [43]
CD11b+Gr-1+	MDSCs inhibit T cell mediated immunoreactivity and GVHD	Decreased number and dysfunction of T cells, the presence of enriched MSCs and/or the increased IL-10, IL-6 cytokine secretion	Morecki et al. [52]
CD11b+Gr-1+	Suppression of acute GVHD by inhibiting alloreactive donor T cell expansion	MDSC suppress GVHD via an IDO-independent manner	Joo et al. [53]
CD11b+Gr-1+	MDSCs suppress allogeneic T cell responses, both in vitro and in vivo	MDSCs triggered arginase-1 activity, which depleted T cell L-arginine	Highfill et al. [54]
CD11bintCD34+	CD34+ M-MDSCs producing NO mediate apoptosis in alloreactive T cell	CD34+ monocytes mobilized with G-CSF require T cell–mediated IFN-γ to yield NO that attenuates T cell activation and proliferation	D’Aveni et al. [55]
CD11b+Gr-1+	MyD88 signaling in donor BM cells demonstrated a protective role via allowing the amplification of MDSCs derived from the donor TCD-BM	GVHD was induced with T cell-depleted BM (TCD-BM) collected from MyD88KO C57BL/6 (B6) mice and T cells collected from WT B6 mice	Lim et al. [56]
M-MDSCs (CD11b+ Ly-6GnegLy-6Chigh) and G-MDSCs (CD11b+Ly-6Gpos Ly-6Clow)	RAPA can significantly alleviate acute graft-versus-host disease	RAPA enhances the immunosuppressive function of PMN-MDSCs via induction of ARG1 and iNOS and stimulation of regulatory T cells in vivo	Lin et al. [57]
M-MDSCs (CD11b+ Ly-6GnegLy-6Chigh) and G-MDSCs (CD11b+Ly-6Gpos Ly-6Clow)	RAPA treatment induced the immunosuppressive role of MDSCs and inhibited GVHD, while GVT effect was maintained	MDSCs from RAPA-treated mice showed increased immunosuppressive potential, which was primarily iNOS-dependent	Scheurer et al. [58]
HLA-DR+CD33+CD14+	Treatment with gal-9 inhibited GVHD	Treatment with gal-9 increased G-MDSCs through stimulations of iNOS and ARG1	Yin et al. [59]
CD11c–CD11b + and Gr-1 +	Infusion of MDSCs and T Regs inhibited aGVHD	Combined treatment modulated differentiation of allogeneic T cells toward T Regs and IL-10-secreting regulatory B cells	Park et al. [60]

HCT, hematopoietic cell transplantation; RAPA, rapamycin; aGVHD, graft-versus-host disease; ARG1, arginase 1; iNOS, inducible nitric oxide synthase.

## Data Availability

Not applicable.

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
