# Peer review of "The Role of Myeloid-Derived Suppressor Cells (MDSCs) in Graft-versus-Host Disease (GVHD)"

_jcm, 2021, doi:10.3390/jcm10102050_

Round 1

Reviewer 1 Report

The role of Myeloid-Derived-Stem-Cells (MDSC) in Graft-Ver-2sus-Host Disease (GVHD) by Demosthenous et al.

General:

1) The term “MDSC” ist used for "myeloid-derived suppressor cells" in the majority of the literature (and also in a part of the manuscript, e.g. in Figure 1 and its legend). Why is the term “myeloid-derived stem cells” used in the rest of the manuscript?  

2) The numbering of the reference list is starting with #66. It seems that the numbering was erroneously initiated (and mixed up) with Table 1.

3) Introduction:

Lines 30-32: “The cornerstone of conventional treatment for aGVHD relies on the use of corticosteroids in combination with other highly immunosuppressive agents, such as anti-thymocyte globulin (ATG)“. – delete ATG (this is well-established and licensed for prevention but not for treatment of GVHD). Instead, it should be noted that ruxolitinib and ibrutinib have been approved by FDA for steroid-refractory aGVHD and cGVHD, respectively. Particularly, ruxolitinib has reached a prominent position among second-line treatment for GVHD because large, radomized studies have shown superiority compared to former „standard oft care“ in aGVHD and cGVHD respectively.  This should be acknowledged.

4) Page 5, lines 122-134. In this paragraph, 2 important stuidies are cited, that revealed an immunosuppressive impact of G-CSF-mobilized blood progenitor cells (Mielcarek et al, 1997; Luyckx et al, 2012). There are two other studies that addressed the impact of progenitor cell mobilization with G-CSF on immune effector cell functions, particularly, in vitro expandability, and revealed the interaction of myeloid progenitor cells with T-lymphocytes responsible for this phenomenon, and should be cited in the given context, since in the era of CAR-T cell therapy and upcoming further cellular therapies including those harnessing NK cells, it is of paramount importance to define optimal conditions for harvesting functionally intact immune effector cells with full proliferative potential.

Author Response

Reviewer 1: We thank reviewer 1 for concise review of our article. We have revised the article and addressed major comments as follows:

  1. The term “MDSC” is used for "myeloid-derived suppressor cells" in the majority of the literature (and also in a part of the manuscript, e.g., in Figure 1 and its legend). Why is the term “myeloid-derived stem cells” used in the rest of the manuscript?

Response: “Myeloid-derived stem cells” has been modified to “Myeloid-derived suppressor cells” which is more suitable.

  1. The numbering of the reference list is starting with #66. It seems that the numbering was erroneously initiated (and mixed up) with Table 1.

Response: The issue has been resolved.

  1. Introduction:

Lines 30-32: “The cornerstone of conventional treatment for aGVHD relies on the use of corticosteroids in combination with other highly immunosuppressive agents, such as anti-thymocyte globulin (ATG)“. – delete ATG (this is well-established and licensed for prevention but not for treatment of GVHD). Instead, it should be noted that ruxolitinib and ibrutinib have been approved by FDA for steroid-refractory aGVHD and cGVHD, respectively. Particularly, ruxolitinib has reached a prominent position among second-line treatment for GVHD because large, radomized studies have shown superiority compared to former „standard oft care“ in aGVHD and cGVHD respectively.  This should be acknowledged.

Response: We agree with the reviewer. The comment regarding ATG has been deleted. Ruxolitinib and ibrutinib had already been referred in the manuscript as alternative, novel biologic agents. However, more emphasis has been given by referring to the studies that were related to (lines 49-54).

  1. Page 5, lines 122-134. In this paragraph, 2 important stuidies are cited, that revealed an immunosuppressive impact of G-CSF-mobilized blood progenitor cells (Mielcarek et al, 1997; Luyckx et al, 2012). There are two other studies that addressed the impact of progenitor cell mobilization with G-CSF on immune effector cell functions, particularly, in vitro expandability, and revealed the interaction of myeloid progenitor cells with T-lymphocytes responsible for this phenomenon, and should be cited in the given context, since in the era of CAR-T cell therapy and upcoming further cellular therapies including those harnessing NK cells, it is of paramount importance to define optimal conditions for harvesting functionally intact immune effector cells with full proliferative potential.

Response: Studies performed by Vedramin et al., Wang et al., and Park et al. addressing the role of G-CSF on immunoregulatory properties of MDSCs have been cited in the context (sections 3.1 and 3.2)

Reviewer 2 Report

In this submission by Demosthenous et al., several comments can be made.

  • Perhaps the first comment is that one has the impression, based on the Title and Abstract, that clinical studies would be discussed. However, the paper is almost exclusively focused on murine studies.  This is due, in part, to most studies being undertaken in rodents and few in humans.  Nonetheless, there are significant clinical observations that speak to GVHD being controlled by MDSCs yet are not discussed by these authors.  Furthermore there are deficiencies in presentation style and discussion that limit enthusiasm for this submission.
  • In the Abstract and elsewhere, the inference is made that GVHD and its limitation by MDSCs is mediated by CD4+ T-cells. Thus, the statement in the Abstract, “In vivo studies demonstrated that MDSCs derived from mobilization protocols can strongly suppress allo-responses mediated by CD4 T-cells and enhance T regulatory activity inhibiting thus GVHD toxicity.”  One can note that MDSCs can limit not only CD4 cells, but also CD8 cells, NK cells and B cells.  One might also comment that CD8 cells are substantially associated with GVHD and have a major role in the mechanism of this pathology.
  • It is confusing, but it appears that the first 65 references are missing in this submission.
  • Despite the inference herein, and as alluded to by some but not all other reviews on this subject, MDSCs have been shown to control GVHD. Thus, the classical comparisons of peripheral blood stem cell transplantation using mobilized stem cell products as compared to bone marrow transplants have documented the importance of the infusion of what we now call MDSCs in the mobilized stem cell product in controlling GVHD.  Thus, transplants using mobilized products infuse approximately 10-fold more T-cells than that found with bone marrow transplant products, yet there is no increase in GVHD in these patients, which has been associated with the infusion of myeloid cells in these products.  Indeed, many investigator initiated protocols with leukemia and lymphoma, accept the role of MDSCs and thus, will infuse T-cell depleted or manipulated products in the absence of MDSCs to retain limited GVL in the absence of extensive GVHD.  Thus, clinicians focused on adoptive cellular therapy understand, and their protocols reflect the presence of MDSCs and their potential role in the regulation of both GVHD and GVL.
  • There are some number of reviews such as the one by Oberholtzer et al. entitled, “Adoptive transfer of regulatory immune cells in organ transplantation” that the authors might find of some interest. It is noted that this paper likely was published after this one was submitted, but the authors may find it an interesting benchmark for consideration.  It is noted that there are other reviews on this subject that they may wish to consider and reference.

Author Response

  1. Perhaps the first comment is that one has the impression, based on the Title and Abstract, that clinical studies would be discussed. However, the paper is almost exclusively focused on murine studies. This is due, in part, to most studies being undertaken in rodents and few in humans.  Nonetheless, there are significant clinical observations that speak to GVHD being controlled by MDSCs yet are not discussed by these authors.  Furthermore, there are deficiencies in presentation style and discussion that limit enthusiasm for this submission.

Response: The section 3.2 referring to studies in humans has been added. Moreover, more studies addressing the implications of MDSCs in GVHD (e.g., from D’Aveni, Lv and Schneidawind and colleagues) have been reviewed and discussion has been updated in an attempt to increase interest and enhance enthusiasm of the manuscript. 

  1. In the Abstract and elsewhere, the inference is made that GVHD and its limitation by MDSCs is mediated by CD4+ T-cells. Thus, the statement in the Abstract, “In vivo studies demonstrated that MDSCs derived from mobilization protocols can strongly suppress allo-responses mediated by CD4 T-cells and enhance T regulatory activity inhibiting thus GVHD toxicity.” One can note that MDSCs can limit not only CD4 cells, but also CD8 cells, NK cells and B cells.  One might also comment that CD8 cells are substantially associated with GVHD and have a major role in the mechanism of this pathology.

Response: We agree with the comment. CD4 T cells were changed to “T cells”, as CD4 and CD8 seem to be affected by MDSCs. Similar modification was performed in the Conclusion part, as well as in the main part (line 116) where B lymphocytes and NK cells were also mentioned.  

  1. It is confusing, but it appears that the first 65 references are missing in this submission.

Response: This issue has been resolved

  1. Despite the inference herein, and as alluded to by some but not all other reviews on this subject, MDSCs have been shown to control GVHD. Thus, the classical comparisons of peripheral blood stem cell transplantation using mobilized stem cell products as compared to bone marrow transplants have documented the importance of the infusion of what we now call MDSCs in the mobilized stem cell product in controlling GVHD. Thus, transplants using mobilized products infuse approximately 10-fold more T-cells than that found with bone marrow transplant products, yet there is no increase in GVHD in these patients, which has been associated with the infusion of myeloid cells in these products.  Indeed, many investigator initiated protocols with leukemia and lymphoma, accept the role of MDSCs and thus, will infuse T-cell depleted or manipulated products in the absence of MDSCs to retain limited GVL in the absence of extensive GVHD.  Thus, clinicians focused on adoptive cellular therapy understand, and their protocols reflect the presence of MDSCs and their potential role in the regulation of both GVHD and GVL.

Response: We thank the reviewer for pointing this out. We have revised the manuscript and a comment has been added in the Conclusion section.

  1. There are some number of reviews such as the one by Oberholtzer et al. entitled, “Adoptive transfer of regulatory immune cells in organ transplantation” that the authors might find of some interest. It is noted that this paper likely was published after this one was submitted, but the authors may find it an interesting benchmark for consideration. It is noted that there are other reviews on this subject that they may wish to consider and reference.

Response: The review from Oberholtzer et al was beneficial and interesting. The study was cited. Other reviews, including those of D’Aveni et al and Law et al have been cited also.

Reviewer 3 Report

The review is timely, topical, and well-written. I felt the authors picked their battles in the sections they chose to discuss, as there's always more you can write, but sometimes it's outside the scope of the document. In this case, they mainly focused on molecular mechanism rather than overall GVHD pathophysiology, which  could potentially be a point of improvement. Therefore, if pressed, I would include the below:   The authors mainly focused on molecular and cellular mechanisms by which MDSCs may inhibit GVHD. However, pathogenesis, outcome, and treatment of GVHD can be very different between organ systems. Therefore, including a section classifying some of the referenced studies by type of GVHD manifestation (IE gastrointestinal, skin, liver) would help to further increase the relevance of this review to clinical translation and better review the available literature.

The authors here review the literature for the role of myeloid-derived stem cells (MDSCs) in the control of GVHD. The writing is generally very good, clear, and concise. However, the document has quite a few typos or other simple errors. These are listed below.

With the exception of these errors, the document is very readable. It excellently catalogues the available literature, and Table 1 in particular (once the table formatting is corrected) will be a very helpful resource for those working in this field in the future. This document needs minor revisions to clean up the document, but otherwise it is very suitable for publication.

Line 15: switch to “thus inhibiting”

Line 19. “translate these findings to a clinical setting.”

Line 41: “antitumour” is incorrect, and please use “α” or “alpha” instead of TNFa

Lines 72-86: Is the font size different here? It looks slightly bigger.

Line 96-97: No comma needed after “nowadays” or “ profiles”.

Line 101: Throughout the paper, “T cell” or “T-cell” is variously used. Please choose one and correct the other usages to fit.

Table 1: All sections of the table are incorrectly inserted as references. Thus, by the bottom of the table, you are beginning your reference list at reference 66. This must be corrected before publication. As it is now, all references listed in the text are unlinked from the bibliography because of this error. Therefore it is difficult to determine what reference numbers point to what references.

Line 156: Use either GvHD or GVHD consistently.

Line 205: “Increasibly” is not a word and should be corrected to “increasingly.”

Line 213: “Involved” is misspelled.

Line 272: “Koehn et colleagues” should be “Koehn and colleagues”

Line 293: Much of section 4 discussed future directions for the work, especially in regards to the use of MDSCs as a therapy. This section touches only lightly on future perspectives, and thus could be amended to “conclusions” alone, or perhaps “discussion”.

Line 307: Acknowledgement section is incompletely filled out

Line 310 – end of document: Reference numbers are incorrect due to Table 1 being listed as references.

Correct throughout: In vivo, in vitro, et. al, and other Latinate terminology should be italicized.

Author Response

Reviewer 3: We thank the reviewer for the constructive comments , helping us to improve the final manuscript. We have addressed the comments as follows:

Response to the comments:

  1. The review is timely, topical, and well-written. I felt the authors picked their battles in the sections they chose to discuss, as there's always more you can write, but sometimes it's outside the scope of the document. In this case, they mainly focused on molecular mechanism rather than overall GVHD pathophysiology, which could potentially be a point of improvement. Therefore, if pressed, I would include the below:   The authors mainly focused on molecular and cellular mechanisms by which MDSCs may inhibit GVHD. However, pathogenesis, outcome, and treatment of GVHD can be very different between organ systems. Therefore, including a section classifying some of the referenced studies by type of GVHD manifestation (IE gastrointestinal, skin, liver) would help to further increase the relevance of this review to clinical translation and better review the available literature.

Response: Unfortunately, the data is limited and little information could be obtained after review of the literature. However, section 3.2 has been added in the document referring to human studies, as its absence was a limitation of the initial draft. In this section we list some studies that involved histopathological analysis and concern humanized GVHD models.

  1. The authors here review the literature for the role of myeloid-derived stem cells (MDSCs) in the control of GVHD. The writing is generally very good, clear, and concise. However, the document has quite a few typos or other simple errors. These are listed below.

With the exception of these errors, the document is very readable. It excellently catalogues the available literature, and Table 1 in particular (once the table formatting is corrected) will be a very helpful resource for those working in this field in the future. This document needs minor revisions to clean up the document, but otherwise it is very suitable for publication.

Typos or simple errors, according to the reviewer

Line 15: switch to “thus inhibiting”   -> Done

Line 19. “translate these findings to a clinical setting.” ->Done

Line 41: “antitumour” is incorrect, and please use “α” or “alpha” instead of TNFa -> Done

Lines 72-86: Is the font size different here? It looks slightly bigger. -> changed

Line 96-97: No comma needed after “nowadays” or “ profiles”. -> modified

Line 101: Throughout the paper, “T cell” or “T-cell” is variously used. Please choose one and correct the other usages to fit. -> changed to “T cell”

Table 1: All sections of the table are incorrectly inserted as references. Thus, by the bottom of the table, you are beginning your reference list at reference 66. This must be corrected before publication. As it is now, all references listed in the text are unlinked from the bibliography because of this error. Therefore it is difficult to determine what reference numbers point to what references. -> corrected

Line 156: Use either GvHD or GVHD consistently. ->  changed to “GVHD”

Line 205: “Increasibly” is not a word and should be corrected to “increasingly.” -> corrected

Line 213: “Involved” is misspelled. -> corrected

Line 272: “Koehn et colleagues” should be “Koehn and colleagues” . -> corrected

Line 293: Much of section 4 discussed future directions for the work, especially in regards to the use of MDSCs as a therapy. This section touches only lightly on future perspectives, and thus could be amended to “conclusions” alone, or perhaps “discussion”. -> We assume the reviewer means section 5. The section has been amended accordingly

Line 307: Acknowledgement section is incompletely filled out -> corrected

Line 310 – end of document: Reference numbers are incorrect due to Table 1 being listed as references. -> corrected

Correct throughout: In vivo, in vitro, et al., and other Latinate terminology should be italicized. -> corrected

Round 2

Reviewer 2 Report

The submission has improved from the original version.  However, insight into GVHD and MDSCs could still be improved. 

The authors are encouraged to review the literature of suppressor cells, especially suppressor cells in stem cell transplantation.  There is a body of literature in the mid-1980s focused on natural suppressor cells (NS) also known as Null cells and other nomenclature that alludes to cellular functions that are now known as MDSCs (A).  Some of these studies in the mid-1990s were directed towards human transplant patients and the stem cell products (B).  This literature is relevant to the impact of myeloid cells on T-cells and their function in cancer patients following bone marrow and hematopoietic stem cell transplantation.  These studies document a significant increase in the numbers of myeloid suppressor cells found in the apheresis products following mobilization, and an inverse relationship between the frequency of these cells and the number of T-cells and T-cell function.  Indeed, there have been a number of studies that have documented an inverse relationship between patient survival and the frequency of these cells in, predominantly, autologous stem cell patients although, to a lesser extent this has been associated with decrease survival in allogeneic stem cell transplant patients (D). 

Early studies regarding Null cells and NS cells as relevant to GVHD have been published by HN Claman and Sam Strober (A).  In addition, the team of Sykes and Sachs undertook a number of interesting murine and canine studies with suppressor cells in allogeneic transplantation.  The authors are directed to all of these studies that go back as early as 1985 (E).  The authors might also find the study by Singh et al., documenting an inverse relationship between suppressor cell activity and CD4 T-cell apoptosis being observed following autologous stem cell transplantation (F).  This study was confirmed shortly afterwards by Mielcarek M in Seattle, which the authors might also find of interest. 

Additional relevant studies have been published by S Sica et al., Korbling et al., L Shao et al. and SE Lee et al., which are more recent clinical studies (G).  Some attention to the role of circulating suppressor cells on the impact of DLIs, as published by LE Franssen et al., might also be considered of relevance to the argument under study (H). 

The point is that there are a significant number of studies that are not addressed, not only in allogeneic stem cell transplantation, but also DLIs and interesting observations that may also be relevant to the argument in autologous stem cell transplantation, particularly those using mobilized stem cell products.  As alluded to in the beginning of this comment, additional understanding and insight into suppressor cell phenotypes, functions, expansion, mobilization and clinical activity would be useful to improve this submission.  Furthermore, as suggested in the Abstract, the impact on GVL activity by these myeloid suppressor cells can be similar to the impact on GVHD and the conclusion that “Concurrently, they do not seem to interfere with GVL effect.”, is argumentative and needs to be substantially supported.

The statement that “The backbone of conventional treatment for aGVHD relies on the use of clinical steroids.”, is somewhat controversial and is disease and therapy dependent. Thus, post-transplant cyclophosphamide in haplo transplants is used to significant effect in preventing the acute GVHD in association with these transplants (I).  One might also note that the N:L ratio can impact clinical outcomes and GVHD and is used clinically for this reason (J). 

The comment “The synergy between T-reg and MDSCs” is not referenced.  If they have a reference it would be good to note the statistical test used to document synergy and the p value for the studies.

One final comment, the authors need to be consistent in calling MDSCs immature or mature.  It is suggested that immature be used, which requires a change to the sentence in “2. Key function of MDSCs”. 

A  Synergism between T and non-T cells in the in vivo induction and in vitro expression of graft-vs.-host disease-induced natural suppressor cells.  Maier T, Holda JH, Claman HN. J Exp Med. 1985 Sep 1;162(3):979-92. doi: 10.1084/jem.162.3.979. PMID: 3161977 Free PMC article.

A T-cell subsets and suppressor cells in human bone marrow.  Schmidt-Wolf IG, Dejbakhsh-Jones S, Ginzton N, Greenberg P, Strober S. Blood. 1992 Dec 15;80(12):3242-50. PMID: 1467527

A Cloned natural suppressor cell lines derived from the spleens of neonatal mice.  Schwadron RB, Gandour DM, Strober S. J Exp Med. 1985 Jul 1;162(1):297-310. doi: 0.1084/jem.162.1.297.

B  Immunosuppressive CD14+HLA-DRlow/neg IDO+ myeloid cells in patients following allogeneic hematopoietic stem cell transplantation.  Mougiakakos D, Jitschin R, von Bahr L, Poschke I, Gary R, Sundberg B, Gerbitz A, Ljungman P, Le Blanc K. Leukemia. 2013 Feb;27(2):377-88. doi: 10.1038/leu.2012.215. Epub 2012 Jul 25. PMID: 22828446

B  Immunologic phenotype and function in human bone marrow, blood stem cells and umbilical cord blood.  Mills KC, Gross TG, Varney ML, Heimann DG, Reed EC, Kessinger A, Talmadge JE. Bone Marrow Transplant. 1996 Jul;18(1):53-61. PMID: 8831996

B  Immunologic attributes of cytokine mobilized peripheral blood stem cells and recovery following transplantation.  Talmadge JE, Reed EC, Kessinger A, Kuszynski CA, Perry GA, Gordy CL, Mills KC, Thomas ML, Pirruccello SJ, Letheby BA, Arneson MA, Jackson JD. Bone Marrow Transplant. 1996 Jan;17(1):101-9.

B Mills KC, Gross TG, Varney ML, Heimann DG, Reed EC, Kessinger A, Talmadge JE. Bone Marrow Transplant. 1996 Jul;18(1):53-61.

D  Predictive Role of Circulating Immune Cell Subtypes Early after Allogeneic Hematopoietic Stem Cell Transplantation in Patients with Acute Leukemia   Tae Woo Kim  1 , Sung-Soo Park  1 , Ji-Young Lim  1 , Gi June Min  1 , Silvia Park  1 , Young-Woo Jeon  1   2 , Seung-Ah Yahng  3 , Seung-Hwan Shin  4 , Sung-Eun Lee  1   2 , Jae-Ho Yoon  1   2 , Byung-Sik Cho  1   2 , Ki-Seong Eom  1   2 , Seok Lee  1   2 , Hee-Je Kim  1   2 , Chang-Ki Min  1   2

E  Natural suppressor cells in spleens of irradiated, bone marrow-reconstituted mice and normal bone marrow: lack of Sca-1 expression and enrichment by depletion of Mac1-positive cells.

Sykes M, Sharabi Y, Sachs DH. Cell Immunol. 1990 May;127(2):260-74. doi: 10.1016/0008-8749(90)90131-a. PMID: 1970279

E  Achieving alloengraftment without graft-versus-host disease: approaches using mixed allogeneic bone marrow transplantation.  Sykes M, Sharabi Y, Sachs DH. Bone Marrow Transplant. 1988 Sep;3(5):379-86. PMID: 3056545 Review.

F  Fas-FasL-mediated CD4+ T-cell apoptosis following stem cell transplantation.   Singh RK, Varney ML, Buyukberber S, Ino K, Ageitos AG, Reed E, Tarantolo S, Talmadge JE. Cancer Res. 1999 Jul 1;59(13):3107-11. PMID: 10397252 Free article. Clinical Trial.

G hG-CSF in healthy donors: mobilization of peripheral hemopoietic progenitors and effect on peripheral blood leukocytes.   Sica S, Rutella S, Di Mario A, Salutari P, Rumi C, Ortu la Barbera E, Etuk B, Menichella G, D'Onofrio G, Leone G. J Hematother. 1996 Aug;5(4):391-7. doi: 10.1089/scd.1.1996.5.391. PMID: 8877714  

G  Delayed effects of rhG-CSF mobilization treatment and apheresis on circulating CD34+ and CD34+ Thy-1dim CD38- progenitor cells, and lymphoid subsets in normal stem cell donors for allogeneic transplantation.   Körbling M, Anderlini P, Durett A, Maadani F, Bojko P, Seong D, Giralt S, Khouri I, Andersson B, Mehra R, vanBesien K, Mirza N, Przepiorka D, Champlin R. Bone Marrow Transplant. 1996 Dec;18(6):1073-9. PMID: 8971375

G  Emerging Role of Myeloid-derived Suppressor Cells in the Biology of Transplantation Tolerance.  Shao L, Pan S, Zhang QP, Jamal M, Rushworth GM, Xiong J, Xiao RJ, Sun JX, Yin Q, Wu YJ, Lie AKW. Transplantation. 2020 Mar;104(3):467-475. doi: 10.1097/TP.0000000000002996. PMID: 31596739 Review.

G  Matrix Metalloproteinase-9 in Monocytic Myeloid-Derived Suppressor Cells Correlate with Early Infections and Clinical Outcomes in Allogeneic Hematopoietic Stem Cell Transplantation.

Lee SE, Lim JY, Kim TW, Jeon YW, Yoon JH, Cho BS, Eom KS, Kim YJ, Kim HJ, Lee S, Cho SG, Kim DW, Lee JW, Min WS, Shin DM, Choi EY, Min CK. Biol Blood Marrow Transplant. 2018 Jan;24(1):32-42. doi: 10.1016/j.bbmt.2017.08.017. Epub 2017 Aug 24. PMID: 28844945

H The impact of circulating suppressor cells in multiple myeloma patients on clinical outcome of DLIs.  Franssen LE, van de Donk NW, Emmelot ME, Roeven MW, Schaap N, Dolstra H, Hobo W, Lokhorst HM, Mutis T. Bone Marrow Transplant. 2015 Jun;50(6):822-8. doi: 10.1038/bmt.2015.48. Epub 2015 Mar 23. PMID: 25798669 Clinical Trial.

I  Posttransplantation cyclophosphamide for prevention of graft-versus-host disease after HLA-matched mobilized blood cell transplantation.  Mielcarek M, Furlong T, O'Donnell PV, Storer BE, McCune JS, Storb R, Carpenter PA, Flowers ME, Appelbaum FR, Martin PJ. Blood. 2016 Mar 17;127(11):1502-8. doi: 10.1182/blood-2015-10-672071. Epub 2016 Jan 13.

J Can Neutrophil-to-Lymphocyte Ratio, Monocyte-to-Lymphocyte Ratio, and Platelet-to-Lymphocyte Ratio at Day +100 be used as a prognostic marker in Multiple Myeloma patients with autologous transplantation?  Serife Solmaz Medeni  1 , Celal Acar  1 , Aybuke Olgun  2 , Alev Acar  2 , Ahmet Seyhanlı  2 , Emin Taskıran  2 , Omur Gokmen Sevindik  3 , Inci Alacacıoglu  2 , Ozden Piskin  2 , Mehmet Ali Ozcan  2 , Fatih Demirkan  2 , Bulent Undar  2 , Guner Hayri Ozsan  2

Author Response

Thank you once again, for giving us the opportunity to submit a revised draft of the review article titled “The role of Myeloid-Derived-Suppressor-Cells (MDSC) in Graft-Versus-Host Disease (GVHD)” to Journal of Clinical Medicine.

Here is a point-by-point response to the reviewers’ comments and concerns. The changes we performed in the manuscript can be recognized as “Tracking” function is on.

Please see below, in blue, our detailed response to comments.

Response to comments:

We thank once again the reviewers for the constructive comments. We have tried to provide a more comprehensive insight on the association of GVHD and MDSCs. Hopefully, we have managed to meet the expectations.

  1. Comments: The authors are encouraged to review the literature of suppressor cells, especially suppressor cells in stem cell transplantation. There is a body of literature in the mid-1980s focused on natural suppressor cells (NS) also known as Null cells and other nomenclature that alludes to cellular functions that are now known as MDSCs (A).  Some of these studies in the mid-1990s were directed towards human transplant patients and the stem cell products (B).  This literature is relevant to the impact of myeloid cells on T-cells and their function in cancer patients following bone marrow and hematopoietic stem cell transplantation.  These studies document a significant increase in the numbers of myeloid suppressor cells found in the apheresis products following mobilization, and an inverse relationship between the frequency of these cells and the number of T-cells and T-cell function.  Indeed, there have been a number of studies that have documented an inverse relationship between patient survival and the frequency of these cells in, predominantly, autologous stem cell patients although, to a lesser extent this has been associated with decrease survival in allogeneic stem cell transplant patients (D).  

Early studies regarding Null cells and NS cells as relevant to GVHD have been published by HN Claman and Sam Strober (A).  In addition, the team of Sykes and Sachs undertook a number of interesting murine and canine studies with suppressor cells in allogeneic transplantation.  The authors are directed to all of these studies that go back as early as 1985 (E).  The authors might also find the study by Singh et al., documenting an inverse relationship between suppressor cell activity and CD4 T-cell apoptosis being observed following autologous stem cell transplantation (F).  This study was confirmed shortly afterwards by Mielcarek M in Seattle, which the authors might also find of interest.

Additional relevant studies have been published by S Sica et al., Korbling et al., L Shao et al. and SE Lee et al., which are more recent clinical studies (G). 

Response: References to null cells have been added according to reviewer’s suggestions. In particular, section 2 (lines 89-93, references #22-26), section 3.1 (lines 166-175, references #22-24, 61-64), section 3.2 (lines 289-291, references #75-76) have been updated.

  1. Comments: Some attention to the role of circulating suppressor cells on the impact of DLIs, as published by LE Franssen et al., might also be considered of relevance to the argument under study (H). The point is that there are a significant number of studies that are not addressed, not only in allogeneic stem cell transplantation, but also DLIs and interesting observations that may also be relevant to the argument in autologous stem cell transplantation, particularly those using mobilized stem cell products. As alluded to in the beginning of this comment, additional understanding and insight into suppressor cell phenotypes, functions, expansion, mobilization and clinical activity would be useful to improve this submission.

Response: The revised version contains additional data, regarding the impact of nuclear suppressor cells and T regs on DLIs (lines 398-405, references #97-98).

  1. Comments: Furthermore, as suggested in the Abstract, the impact on GVL activity by these myeloid suppressor cells can be similar to the impact on GVHD and the conclusion that “Concurrently, they do not seem to interfere with GVL effect.”, is argumentative and needs to be substantially supported.

Response: The reviewer is right. The statements have been modified to “However, the influence of MDSCs on GVL effect is not fully defined.”, lines 16-17.

  1. Comments: The statement that “The backbone of conventional treatment for aGVHD relies on the use of clinical steroids.”, is somewhat controversial and is disease and therapy dependent. Thus, post-transplant cyclophosphamide in haplo transplants is used to significant effect in preventing the acute GVHD in association with these transplants (I). One might also note that the N:L ratio can impact clinical outcomes and GVHD and is used clinically for this reason (J).

Response: GVHD mortality and morbidity are high. Comments regarding GVHD prophylaxis have been added in the Introduction, lines 58-60. References to EBMT guidelines and post-transplant cyclophosphamide have also been added (references #10-11).

  1. Comments: The comment “The synergy between T-reg and MDSCs” is not referenced. If they have a reference it would be good to note the statistical test used to document synergy and the p value for the studies.

Response: p value and statistical test have been added according to reviewer’s comments. Moreover, an additional interesting study has been added about the role of T regs on MDSC differentiation (reference #96).

  1. Comments: One final comment, the authors need to be consistent in calling MDSCs immature or mature. It is suggested that immature be used, which requires a change to the sentence in “2. Key function of MDSCs”.

Response: MDSCs are characterized as immature throughout the manuscript.

Sincerely yours,

The authors